# Improvement of liver fibrosis, but not steatosis, after HCV eradication as assessment by MR-based imaging: Role of metabolic derangement and host genetic variants

**Natthaya Chuaypen[1], Surachate Siripongsakun[2], Pantajaree Hiranrat[2], Natthaporn Tanpowpong[3], Anchalee Avihingsanon[4], Pisit Tangkijvanich**[ID][1] *

1 Center of Excellence in Hepatitis and Liver Cancer, Department of Biochemistry, Faculty of Medicine, Chulalongkorn University, Bangkok, Thailand, 2 Sonographer School, Faculty of Health Science Technology, Chulabhorn Royal Academy, Bangkok, Thailand, 3 Department of Radiology, Faculty of Medicine, Chulalongkorn University, Bangkok, Thailand, 4 The HIV Netherlands Australia Thailand Research Collaboration (HIV NAT), Bangkok, Thailand

* pisittkvn@yahoo.com

**Data Availability Statement:** All relevant data are within the paper.

## Abstract

Significant liver fibrosis regression occurs after hepatitis C virus (HCV) therapy. However, the impact of direct-acting antivirals (DAAs) on steatosis is less clear. This study was aimed at evaluating serial fibrosis and steatosis alterations in patients with HCV genotype 1, who achieved sustained virological response (SVR). We enrolled 55 HCV mono-infected and 28 HCV/HIV co-infected patients receiving elbasvir/grazoprevir from a clinical trial. Fibrosis and steatosis were assessed at baseline, follow-up week-24 (FUw24) and week-72 (FUw72) by magnetic resonance elastography (MRE) and proton density fat fraction (PDFF), respectively. Patatin-like phospholipase domain-containing protein 3 (*PNPLA3*) rs738409, transmembrane six superfamily member 2 (*TM6SF2*) rs58542926 and membrane bound O-acyltransferase domain-containing 7 (*MBOAT7*) rs641738 polymorphisms were determined by allelic discrimination. Overall, mean MRE decreased significantly from baseline to FUw24 and FUw72. At FUw72, patients with baseline F2-F4 had higher rate of ≥30% MRE decline compared with individuals with baseline F0-F1 (30.2%vs.3.3%, *P* = 0.004). In multivariate analysis, significant fibrosis was associated with MRE reduction. The prevalence of steatosis (PDFF≥5.2%) at baseline was 21.7%. Compared to baseline, there were 17 (20.5%) patients with decreased PDFF values at FUw72 (<30%), while 23 (27.7%) patients had increased PDFF values (≥30%). Regarding the overall cohort, mean PDFF significantly increased from baseline to FUw72, and displayed positive correlation with body mass index (BMI) alteration. In multivariate analysis, the presence of diabetes, *PNPLA3* CG+GG genotypes and increased BMI at FUw72 were significantly associated with progressive steatosis after SVR. Other genetic variants were not related to fibrosis and steatosis alteration. This study concluded that HCV eradication was associated with fibrosis improvement. However, progressive steatosis was observed in a proportion of patients, particularly among

**Funding:** PT was supported by The Thailand Research Fund (TRF) Senior Research Scholar (RTA6280004). NC received Ratchadapisek Sompoch Endowment Fund (DNS 64_007_30_002_1), Chulalongkorn University and Ratchadaphiseksomphot Matching Fund (RA-MF-22/64), Faculty of Medicine, Chulalongkorn University. The funders had no role in study design, data collection and analysis, decision to publish, or preparation of the manuscript.

**Competing interests:** The authors have declared that no competing interests exist.

individuals with metabolic derangement and *PNPLA3* variants. The combined clinical parameters and host genetic factors might allow a better individualized strategy in this sub-group of patients to alleviate progressive steatosis after HCV cure.

## Introduction

Hepatitis C virus (HCV) infection represents a global public health problem. The current estimated prevalence of chronic HCV infection is 1.0%, accounting for 70 million people worldwide [1]. It has been shown that 10–20% of chronically infected patients could develop long-term complications including cirrhosis and hepatocellular carcinoma (HCC) [2]. Additionally, there are 2.3 million people with human immunodeficiency virus (HIV) co-infected with HCV, of whom 80% are people who inject drugs [3]. Comparing with HCV mono-infection, the risk of liver-related complications is significantly greater in HCV/HIV co-infection [4]. As a result of effective direct-acting antivirals (DAAs), HCV mono- and HCV/HIV co-infected patients can now achieve sustained virological response (SVR) rates over 95% [5]. In line with these data, our recent report demonstrated that the combination of elbasvir and grazoprevir (EBR/GZR) was effective for HCV-genotype 1 (GT1)-infected Thai patients with or without HIV infection, as the overall SVR12 and SVR24 rates were 98.0% and 95.0%, respectively [6].

Collecting data reveal that HCV clearance is associated with regression of liver fibrosis, which leads to reduced risk of cirrhosis and HCC development [5]. Thus, an accurate evaluation of dynamic changes in liver fibrosis is needed for the monitoring of patients treated with DAAs. Although liver biopsy is the gold standard for the measurement of liver histopathology, this invasive procedure is associated with complications and is prone to sampling error; thus, is rarely performed after SVR [7]. At present, magnetic resonance elastography (MRE), a magnetic resonance imaging (MRI)-based technique for tissue stiffness quantification, is the most accuracy non-invasive method for liver fibrosis [8, 9]. Additionally, MRI-based proton density fat fraction (PDFF) has emerged as a reliable alternative to liver biopsy in detecting liver steatosis [10]. Overall, the assessment of fibrosis and steatosis by these MRI-based methods are better than vibration-controlled transient elastography (VCTE) and controlled attenuation parameter (CAP), respectively [11].

Although reducing in fibrosis is feasible after SVR, it is unclear whether HCV clearance might induce steatosis improvement. In this context, some studies demonstrated a reduction in liver steatosis after SVR [12–14], while other reports showed a tendency towards continuing or increased steatosis from baseline [15–18]. These conflicting results emphasize the need for further studies with a longer duration of follow-up that could identify factors predictive of steatosis change in patients achieved SVR. Indeed, recent data have suggested that steatosis is an independent risk factor for HCC after HCV eradication [19]. Growing evidence has also revealed that host genetic factors play an essential role in clinical outcome of chronic HCV infection [20]. For instance, several single nucleotide polymorphisms (SNPs) such as patatin-like phospholipase domain-containing protein 3 (*PNPLA3*), transmembrane six superfamily member 2 (*TM6SF2*) and membrane bound O-acyltransferase domain-containing 7 (*MBOAT7*) appear to be associated with steatosis progression [21]. Whether these SNPs are related to developing steatosis after SVR remains unclear and needs further investigation.

To address this important issue, this prospective study was designed to investigate serial changes of MRE and PDFF values in patients with HCV and HIV/HCV infection after EBR/

GZR therapy. We also assessed the potential role of steatosis-related SNPs associated with treatment outcome in these patients.

## Materials and methods

### Patients

This non-randomized, open-label prospective cohort was conducted in King Chulalongkorn Memorial Hospital, Bangkok, Thailand. Between August 2018 and June 2019, 101 patients with HCV GT1 with or without HIV infection were recruited and treated with EBR/GZR (clinicaltrials.gov: NCT03037151) [6]. Inclusion criteria were individuals aged≥18 years with confirmed chronic HCV infection by anti-HCV positivity>6 months and serum HCV RNA>10,000 IU/mL. Among HCV/HIV co-infection, each patient had undetectable plasma HIV-RNA levels during antiretroviral therapy (ART) at enrollment. Exclusion criteria were concomitant hepatitis B virus (HBV) infection, presence of other liver diseases (e.g., alcohol liver disease, autoimmune hepatitis and Wilson's disease), previous DAA therapy and evidence of decompensated cirrhosis or HCC. Treatment-naïve patients were treated with EBR/GZR for 12 weeks, while the treatment-experienced group with pegylated interferon (PEG-IFN) and ribavirin (RBV) received EBR/GZR plus weight-based RBV for 16 weeks. After therapy, SVR (defined by HCV RNA level <12 IU/mL) at week 12 (SVR12) and week 24 (SVR24) were assessed. In addition, these patients had long-term followed-up at week 72 after treatment completion.

The study protocol was approved by the Institutional Review Board (Faculty of Medicine, Chulalongkorn University, Bangkok, Thailand; IRB No.483/59). Written informed consent was obtained from all participants for their clinical information and specimens.

### Laboratory assays

Serum HCV RNA was measured by real-time quantitative reverse-transcription polymerase chain reaction (RT-PCR) (Abbott Molecular Inc. Des Plaines, IL, USA). HCV genotypes were determined by nucleotide sequencing of the core and NS5B regions as previously described [22]. Plasma HIV RNA was assessed by the Abbott RealTime HIV-1 Assay (Abbott Molecular Inc. Des Plaines, IL, USA).

### DNA preparation and genetic analysis

Genomic DNA was extracted from 100 μl peripheral blood mononuclear cells (PBMCs) using phenol-chloroform-isoamyl alcohol isolation method [23]. The quality of DNA was measured using spectrophotometer (NanoDrop 2000c, Thermo Scientific). The respective SNPs, including *PNPLA3* rs738409, *TM6SF2* rs58542926 and *MBOAT7* rs641738, were genotyped using real-time PCR protocol based on *Taq*Man assays. The reaction was performed including 4 μl of 2.5X master mix (5 PRIME, Germany), 0.25 μl of 40X primers and probes mixture *Taq*Man SNP Genotyping Assay (assay ID:C_7241_10), Applied Biosystems, USA), 50–100 ng of genomic DNA and nuclease-free water to the final volume of 10 μl. The real-time PCR condition was performed in StepOne Plus Real-time PCR system (Applied Biosystems, USA) according to the manufacturer's protocol. Briefly, initial denaturation was hold at 95˚C for 10 min, followed by 50 cycles of amplification including denaturation at 92˚C for 10 sec, and annealing/extension at 60˚C for 1 min. Fluorescent signals (FAM and VIC) were acquired at the end of each cycle. Positive and negative controls were included in each experiment to confirm data interpretation. Allelic discrimination plot was analyzed using StepOne TM software (version 2.2, Applied Biosystems).

## Assessment of liver stiffness and steatosis

MRE and PDFF was performed at baseline prior to DAAs, at 24 weeks and 72 weeks of follow-up (FU) by MR imaging system Philips Ingenia at 3.0 T (Philips Healthcare, Best, the Netherlands) as previously described [24, 25]. Based on data regarding chronic HCV infection, the cut-off values for fibrosis ≥F2, ≥F3 and F4 were 3.2, 4.0 and 4.6 kPa, respectively [26]. Improvement of liver stiffness at FUw72 was defined by ≥30% decrease of MRE values from baseline. For PDFF, the presence of steatosis (affecting ≥5% of hepatocytes) was defined as value ≥5.2%. Additionally, the cut-off values of PDFF for diagnosing steatosis grades ≥1, ≥2 and ≥3 were 5.2%, 11.3% and 17.1% respectively [27]. Progressive steatosis at FUw72 was characterized by ≥30% increase of PDFF values from baseline.

## Statistical analyses

Data were showed as percentages or mean ± standard deviation (SD). Comparisons between groups were analyzed by $\chi^2$ or Fisher's exact test for categorical variables and by two-sample t tests for continuous variables. Alteration of parameters during follow-up were calculated by repeated measurement analysis with baseline data as covariate, using a generalized linear mixed models (GLMMs) and Bonferroni correction for multiple comparisons [28, 29]. Correlations between parameters were analyzed by Spearman's rank test. To test whether the SNPs deviation from Hardy-Weinberg equilibrium, $\chi^2$ test was calculated as described previously [30]. Uni- and multi-variate analysis were performed to calculate parameters associated with MRE and PDFF alterations. $P$ value<0.05 was considered as a statistical significance. The data analyses were performed by IBM SPSS software for Windows version 23.0 (IBM, Chicago, IL, USA).

## Results

### Baseline clinical characteristics

Among total 101 treated cases, 83 (82.2%) patients achieved SVR and had serial MRI-based assessment (baseline, EOT-wk24 and EOT-wk72) were recruited in this study. Table 1 demonstrates baseline characteristics regarding HIV status. There were 64 (77.1%) males and 19 (22.9%) females with their mean age of 47.9±10.2 years and there were 55 (66.3%) and 28 (33.7%) individuals with or without HIV infection, respectively. Compared to the HCV/HIV group, the HCV group had higher mean age, the frequency of HCV GT1b and MRE value. In contrast, the co-infected group had significantly higher serum HCV RNA level than the mono-infected group. There was no significant difference between groups regarding body mass index (BMI), diabetes, biochemical parameters, PDFF value and previous HCV therapy, as well as the distribution of *PNPLA3* rs738409, *TM6SF2* rs58542926 and *MBOAT7* rs641738 genotypes.

### Fibrosis at baseline and during follow-up

At baseline, MRE value was correlated with age (r = 0.411, $P$<0.001), AST (r = 0.573, $P$<0.001), ALT (r = 0.391, $P$<0.001) and PDFF value (r = 0.262, $P$ = 0.017). A negative correlation was found between MRE and platelet counts (r = -0.577, $P$<0.001). There was no correlation between MRE and other clinical parameters such as BMI and HCV RNA level.

Fibrosis stages at baseline, FUw24 and FUw72 are shown in Fig 1A. At baseline, there were 30(36.1%), 27(32.5%), 13(15.7%) and 13(15.7%) patients with F0-F1, F2, F3 and F4, respectively. At FUw24, the corresponding numbers were 35(42.2%), 33(39.8%), 9(10.8%) and 6 (7.2%), respectively, while the corresponding numbers at FUw72 were 50(60.2%), 22(26.5%), 7

**Table 1. Baseline characteristics of patients in this study.**

| Baseline Characteristics | HCV mono-infection (n = 55) | HCV/HIV co-infection (n = 28) | P |
|---|---|---|---|
| Age (years) | 50.9±10.0 | 42.1±8.0 | <0.001* |
| Gender | | | 0.583 |
| Male | 41(74.5) | 23(82.1) | |
| Female | 14(25.5) | 5(17.9) | |
| Body mass index (kg/m$^2$) | | | 0.188 |
| <25 | 37(67.3) | 24(85.7) | |
| 25–30 | 15(27.3) | 3(10.7) | |
| >30 | 3(5.4) | 1(3.6) | |
| Diabetes | 10(18.2) | 3(10.7) | 0.528 |
| Aspartate aminotransferase (IU/L) | 55.1±38.9 | 43.4±16.6 | 0.133 |
| Alanine aminotransferase (IU/L) | 64.5±53.3 | 54.5±26.9 | 0.261 |
| Platelet count (10$^9$/L) | 189.2±71.5 | 210.7±66.9 | 0.189 |
| Log$_{10}$ HCV RNA (IU/mL) | 6.2±0.7 | 6.5±0.6 | 0.039* |
| HCV genotype | | | 0.043* |
| GT1a | 35(63.6) | 24(85.7) | |
| GT1b | 20(36.4) | 4(14.3) | |
| Magnetic resonance elastography (MRE, kPa) | 3.5±1.1 | 3.0±0.7 | 0.032* |
| Proton density fat fraction (PDFF, %) | 3.8±2.5 | 4.4±3.66 | 0.371 |
| *PNPLA3* rs738409 | | | 1.000 |
| CC | 30(54.5) | 15(53.6) | |
| CG+GG | 25(45.5) | 13(46.4) | |
| *TM6SF2* rs58542926 | | | 0.240 |
| CC | 42(76.4) | 25(89.3) | |
| CT+TT | 13(23.6) | 3(10.7) | |
| *MBOAT7* rs641738 | | | 0.350 |
| CC | 27(49.1) | 10(35.7) | |
| CT+TT | 28(50.9) | 18(64.3) | |
| Previous PEG-IFN/RBV therapy | 14(25.5) | 6(21.4) | 0.790 |

Data express as mean ± SD or n (%)

*P-value<0.05

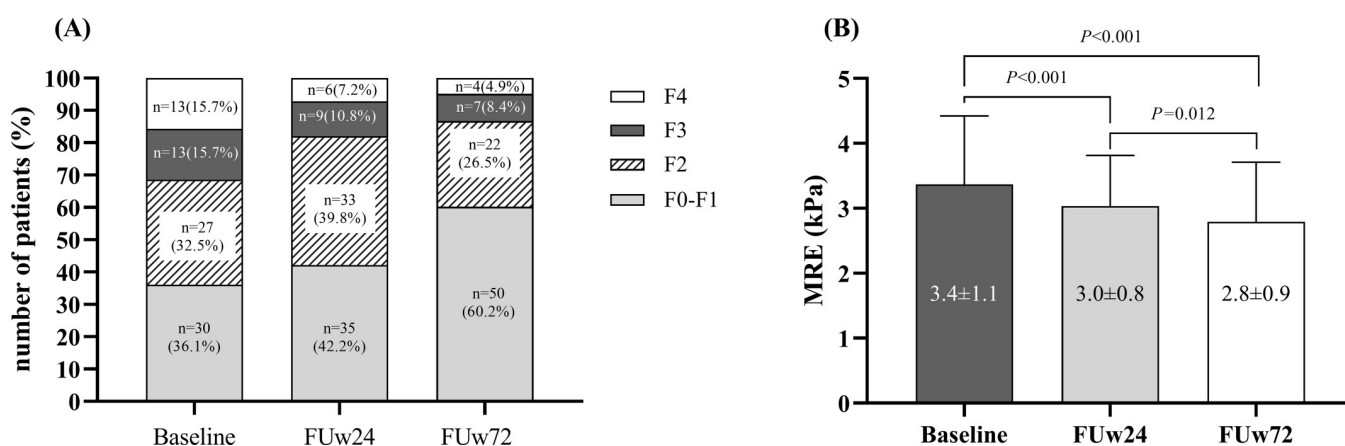

**Fig 1.** Changes of liver stiffness at baseline, FUw24 and FUw72 (A) Liver stiffness stages (B) Mean liver stiffness.

(8.4%) and 4(4.9%), respectively. In the overall cohort, MRE values significantly decreased from baseline to FUw24 (3.4±1.1 vs 3.0±0.8 kPa, $P$ = 0.001), from FUw24 to FUw72 (3.0±0.8 vs 2.8±0.9 kPa, $P$ = 0.036) and from baseline to FUw72 ($P$<0.001) (Fig 1B). There were 17 (20.5%) patients with decreased MRE values at FUw72 (≥30% compared with baseline). Notably, patients with baseline F2-F4 had significantly higher MRE decline (≥30%) rate than individuals with baseline F0-F1 (30.2% vs. 3.3%, $P$ = 0.004).

### Steatosis at baseline and during follow-up

At baseline, PDFF value was correlated with BMI (r = 0.432, $P$<0.001), AST (r = 0.380, $P$<0.001), ALT (r = 0.439, $P$<0.001) and MRE value (r = 0.262, $P$ = 0.017). There was no correlation between PDFF and other clinical parameters including age, platelet count and HCV RNA level. During follow-up, there were no correlation between steatosis and fibrosis at the same time points (FUw24; r = 0.006, $P$ = 0.995 and FUw72; r = 0.101, $P$ = 0.362).

Based on PDFF ≥5.2%, there were 18 (21.7%),14 (16.9%) and 29 (34.9%) patients with steatosis at baseline, FUw24 and FUw72, respectively. At baseline, there were 65(78.3%), 14 (16.9%) and 4(4.8%) patients with grade 0, 1 and 2 steatosis, respectively. At FUw24, there were 69(83.1%), 7(8.4%), 5(6.0%) and 2(2.5%) patients with grade 0, 1, 2 and 3 steatosis, respectively, while the corresponding number at FUw72 were 54(65.1%), 15(18.1%), 7(8.4%) and 7(8.4%), respectively (Fig 2A). In the overall cohort, mean PDFF values did not significantly differ between baseline and FUw24 (4.0±2.9% vs 3.8±3.9%, $P$ = 1.000). However, mean PDFF increased significantly from FUw24 to FUw72 (3.8±3.9% vs 5.8±6.6%, $P$ = 0.004) and from baseline to FUw72 (4.0±2.9% vs 5.8±6.6%, $P$ = 0.015) (Fig 2B).

Among individuals with baseline steatosis, a resolution of liver fat (PDFF<5.2%) at FUw72 was observed in 2 (11.1%) patients, while the remaining cases had persistent or increased steatosis. In patients without baseline steatosis, clinical fatty liver (PDFF ≥5.2%) was detected in 13 (20%) patients at FUw72. Compared to baseline, there were 17 (20.5%) patients with decreased PDFF values at FUw72 (<30%), while 23 (27.7%) patients had increased PDFF values (≥30%). Interestingly, there was a positive correlation (r = 0.307, $P$ = 0.005) between PDFF and BMI change at FUw72 compared with baseline.

Regarding BMI at each timepoint, our data showed that there were no significant changes between BMI at baseline and FUw24 (23.6±3.8 vs 23.7±3.8 kg/m$^2$, $P$ = 0.909) and between FUw24 and FUw72 (23.7±3.8 vs 24.0±3.7 kg/m$^2$, $P$ = 0.088). However, the average BMI at FUw72 (24.0±3.7 kg/m$^2$) was significantly higher when compared with baseline ($P$ = 0.039).

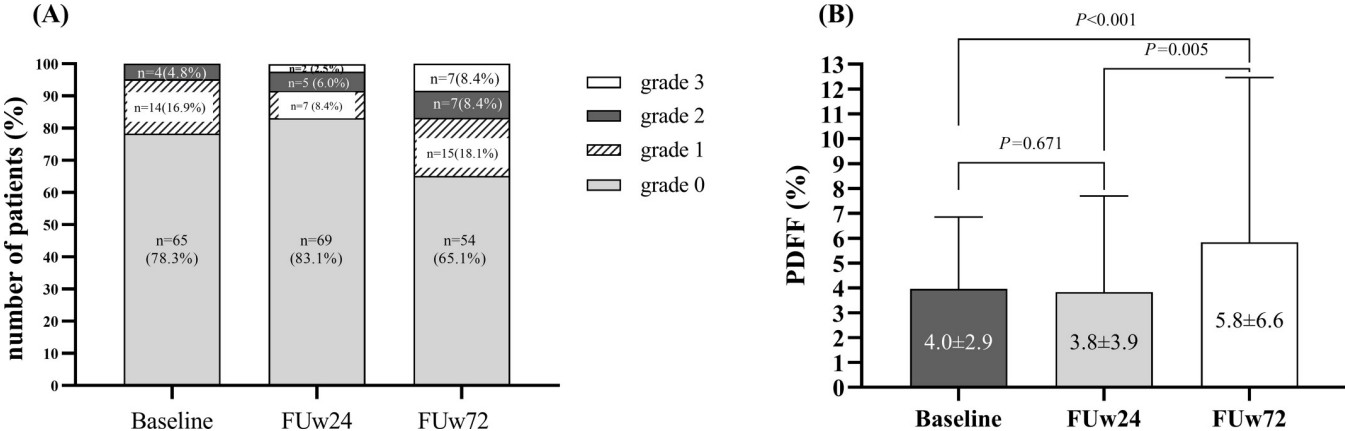

**Fig 2.** Changes of liver steatosis at baseline, FUw24 and FUw72 (A) Liver steatosis grades (B) Mean liver steatosis.

To compare the alterations of fibrosis and steatosis regarding treatment outcome, serial MRE and PDFF at each time point were compared between 83 patients with SVR and 4 patients without SVR. These non-responders were participants enrolled from the same cohort [6] including 3 (75%) males and one (25%) female with their mean age of 42.3±10.8 years. As shown in S1 Fig, patients with SVR, compared to those without SVR, had comparable MRE values at baseline and FUw24 but had significantly declined levels at FUw72. For serial PDFF values, there was no significant difference between the two groups at baseline, FUw24 and FUw72.

## Distributions of SNPs

The genotyping of the SNPs was successfully performed in all samples. In addition, all tested SNPs were in Hardy-Weinberg equilibrium. The distribution of *PNPLA3* rs738409 CC, CG and GG genotypes was 45(54.2%), 26(31.3%) and 12(14.5%), respectively. For *TM6SF2* rs58542926, the frequency of CC, CT and TT genotype was 67(80.7%), 13(15.7%) and 3(3.6%), respectively, while the distribution of *MBOAT7* rs641738 CC, CT and TT genotypes was 37 (44.6%), 35(42.2%) and 11(13.3%), respectively.

At baseline, there was no difference in the frequency of steatosis between patients carried *PNPLA3* CC and CG+GG [8(17.8%) vs. 10(26.3%), *P* = 0.426]. Likewise, patients harboring *TM6SF2* CC had similar rate of steatosis compared to those with CT+TT [16(23.9%) vs. 2 (12.5%), *P* = 0.502]. Additionally, patients carried *MBOAT7* CC and CT+TT had comparable rate of steatosis [8(21.6%) vs. 10(21.7%), *P* = 1.000].

At FUw72, patients carried *PNPLA3* CG+GG tended to have higher frequency of steatosis compared to those with CC genotype, although the significance was not reached [17(44.7%) vs. 12(26.7%), *P* = 0.108]. Regarding *TM6SF2*, patients harboring CC and CT+TT genotypes had similar rate of steatosis [23(34.3%) vs. 6(37.5%), *P* = 1.000]. Similarly, the rate of baseline steatosis in patients carried *MBOAT7* CC and CT+TT was comparable [12(32.4%) vs. 17 (37.0%), *P* = 817]. Considering increased PDFF≥30% from baseline, its frequency was significantly observed in patients carried *PNPLA3* CG+GG than those with CC genotype [16(42.1%) vs. 7(15.6%), *P* = 0.013]. For *TM6SF2* and *MBOAT7* genotypes, there was no difference between the CC and non-CC groups regarding increased PDFF values at FUw72 [17(25.4%) vs. 6 (37.5%), *P* = 0.360 and 10(27.0%) vs. 13(28.3%), *P* = 1.000, respectively).

## Factors associated with the alteration of MRE and PDFF values

Univariate and multivariate analyses were calculated to identify baseline factors associated with decreased MRE (≥30% from baseline) at FUw72. These factors included age, gender, BMI, diabetes, HIV status, previous treatment, ALT, platelet counts, HCV sub-genotype, HCV RNA level, fibrosis staging, steatosis grading and the SNP genotypes, as well as BMI change at FUw72. In univariate analysis, parameters associated with decreased MRE were low platelet counts and significant fibrosis (≥F2). In multivariate analysis, only significant fibrosis was independent factor associated with MRE reduction (Table 2).

Similar baseline parameters were included for the analysis of factors associated with increased PDFF (≥30% from baseline) at FUw72. In univariate and multivariate analyses, the presence of diabetes and *PNPLA3* CG+GG genotypes and increased BMI at FUw72 were significantly associated with progressive steatosis after SVR (Table 3). As percentage of BMI change (≥5% from baseline) was considered to be an independent factor associated with an increased PDFF, we further evaluated repeated variables effect on the change in PDFF value using the generalized linear mixed effects models. In this context, our result demonstrated that BMI in separate time-points were not associated with PDFF alteration overtime (*P* = 0.136).

**Table 2. Factors associated with MRE reduction (≤30% from baseline).**

| Factors | Category | Univariate analysis | | Multivariate analysis | |
|---|---|---|---|---|---|
| | | OR (95%CI) | *P* | OR (95%CI) | *P* |
| Baseline | | | | | |
| Age (years) | ≥ 45 vs. < 45 | 1.53 (0.51–4.62) | 0.453 | | |
| Gender | Male vs. Female | 0.67 (0.17–2.63) | 0.566 | | |
| BMI (kg/m$^2$) | ≥ 25 vs. < 25 | 0.82 (0.24–2.85) | 0.755 | | |
| Diabetes | Yes vs. No | 1.95 (0.52–7.32) | 0.323 | | |
| HIV positivity | No vs. Yes | 1.50 (0.50–4.49) | 0.468 | | |
| Previous HCV therapy | Yes vs. No | 2.03 (0.64–6.44) | 0.232 | | |
| Alanine aminotransferase (IU/L) | ≥ 60 vs. < 60 | 0.96 (0.31–2.91) | 0.935 | | |
| Platelet count (10$^9$/L) | < 175 vs. ≥ 175 | 3.21 (1.05–9.78) | 0.040* | 1.72 (0.52–5.69) | 0.372 |
| HCV sub-genotype | GT1b vs. GT1a | 1.03 (0.32–3.33) | 0.960 | | |
| Log$_{10}$ HCV RNA (IU/mL) | ≥ 6.0 vs. < 6.0 | 2.07 (0.54–8.03) | 0.291 | | |
| Liver fibrosis staging | F2-F4 vs. F0-F1 | 12.54 (1.57–100.17) | 0.017* | 9.84 (1.15–84.05) | 0.037* |
| Liver steatosis | Yes vs. No | 0.80 (0.20–3.26) | 0.760 | | |
| *PNPLA3* rs738409 | CG+GG vs. CC | 1.94 (0.60–5.72) | 0.231 | | |
| *TM6SF2* rs58542926 | CT+TT vs. CC | 1.39 (0.38–5.00) | 0.619 | | |
| *MBOAT7* rs641738 | CT+TT vs. CC | 0.88 (0.30–2.57) | 0.818 | | |
| FUw72 | | | | | |
| BMI (percentage change from baseline) | ≥ 5.0 vs. <5.0 | 1.19 (0.40–3.51) | 0.752 | | |

OR, odd ratio; CI, confident interval

*P-value<0.05

**Table 3. Factors associated with increased PDFF (≥30% from baseline).**

| Factors | Category | Univariate analysis | | Multivariate analysis | |
|---|---|---|---|---|---|
| | | OR (95%CI) | *P* | OR (95%CI) | *P* |
| Baseline | | | | | |
| Age (years) | ≥ 45 vs. < 45 | 1.27 (0.48–3.39) | 0.629 | | |
| Gender | Male vs. Female | 0.91 (0.29–2.90) | 0.877 | | |
| BMI (kg/m$^2$) | ≥ 25 vs. < 25 | 1.31 (0.45–3.80) | 0.616 | | |
| Diabetes | Yes vs. No | 5.87 (1.67–20.57) | 0.006* | 6.72 (1.62–27.83) | 0.009* |
| HIV positivity | No vs. Yes | 1.39 (0.51–3.77) | 0.521 | | |
| Previous HCV therapy | Yes vs. No | 1.58 (0.54–4.66) | 0.405 | | |
| Alanine aminotransferase (IU/L) | ≥ 60 vs. < 60 | 0.92 (0.34–2.52) | 0.873 | | |
| Platelet count (10$^9$/L) | < 175 vs. ≥ 175 | 1.08 (0.41–2.84) | 0.881 | | |
| HCV sub-genotype | GT1b vs. GT1a | 1.47 (0.52–4.11) | 0.405 | | |
| Log$_{10}$ HCV RNA (IU/mL) | ≥ 6.0 vs. < 6.0 | 1.58 (0.51–4.92) | 0.429 | | |
| Liver fibrosis staging | F2-F4 vs. F0-F1 | 1.09 (0.40–2.97) | 0.873 | | |
| Liver steatosis | Yes vs. No | 1.95 (0.65–5.87) | 0.236 | | |
| *PNPLA3* rs738409 | CG+GG vs. CC | 3.95 (1.41–11.08) | 0.009* | 3.80 (1.22–11.79) | 0.021* |
| *TM6SF2* rs58542926 | CT+TT vs. CC | 1.77 (0.56–5.59) | 0.334 | | |
| *MBOAT7* rs641738 | CT+TT vs. CC | 1.06 (0.40–2.80) | 0.901 | | |
| FUw72 | | | | | |
| BMI (percentage change from baseline) | ≥ 5.0 vs. <5.0 | 3.58 (1.30–9.88) | 0.014* | 4.48 (1.40–14.32) | 0.011* |

OR, odd ratio; CI, confident interval

*P-value<0.05.

## Discussion

Non-invasive measurement of longitudinal changes in fibrosis and steatosis in patients undergoing DAA therapy is an essential unmet need. At present, MRI-based modality is the most accurate non-invasive technique for fibrosis and steatosis determination [8, 11]. Compared to VCTE and CAP, MRE and PDFF have an advantage of visualizing the entire liver that could diminish sampling variability from the heterogeneous distribution of fibrosis and steatosis, respectively. In a recent meta-analysis, MRE displays high area under the ROC curves (AUROCs) of approximately 0.90 in determining fibrosis ≥F2, ≥F3 and cirrhosis [31]. Additionally, PDFF has an excellent accuracy for detecting steatosis in nonalcoholic fatty liver disease (NAFLD), as its AUROCs for categorizing steatosis grades 0 vs. 1–3, 0–1 vs. 2–3, and 0–2 vs. 3 are 0.98, 0.91, and 0.90, respectively [10]. Indeed, PDFF is a more practical option for quantifying dynamic changes in liver fat content compared to histological studies and has been used as a primary endpoint in several NAFLD trials [32, 33]. Moreover, emerging data indicate that a 30% relative alteration in PDFF is associated with significant histological changes and represents a potential surrogate marker for evaluating treatment outcome in clinical trials [34]. Similarly, a threshold of MRE reduction of ≥30%, could be used as an appropriate alternative to liver biopsy with a high specificity of 100% [35]. Thus, in this study we applied the cut-off 30% relative changes in MRE or PDFF values as the endpoints at FUw72 for quantifying dynamic changes of liver fibrosis or steatosis, respectively.

Currently, longitudinal assessment of stiffness based on MRE in patients treated with DAAs is inadequate. In this study, we demonstrated significantly decreasing stiffness from baseline to FUw24 in patients achieved SVR, which were in line with previous reports using MRE as the reference [12, 36]. These findings suggest that successful DAAs could lead to stiffness improvement within months after therapy. However, it is still unclear whether short-term stiffness improvement might indicate real fibrosis reduction or only the resolution of necroinflammatory activities [37]. To this end, our results provided additional evidence indicating that MRE significantly declined throughout the long-term follow-up (FUw72), which might reflect true and continuing fibrosis regression after SVR. In this report, approximately 40% of responders with baseline F3-F4 had fibrosis regression ≥30% on MRE. This finding was in line with previous data demonstrating that HCV eradication could yield a regression of advanced fibrosis/cirrhosis but the results might differ substantially among individuals [38].

The prevalence of steatosis is common in HCV-GT3 as this genotype is able to induce steatosis by direct and indirect mechanisms [39]. Earlier reports revealed that HCV eradication by IFN-based therapy attenuated steatosis in HCV-GT3, supporting the role of virus-related steatosis [39, 40]. In this report, approximately 20% of patients had high baseline PDFF, suggesting significant steatosis was not uncommon in HCV-GT1. Notably, baseline steatosis in our study was found predominantly in patients with higher BMI and diabetes, which was similar to previous data [41]. In this study, the frequencies of overweight and obesity were 20% and 5%, respectively in patients with high baseline PDFF, suggesting that steatosis frequently occurred in non-obese Asian populations [42]. Given the mechanism of steatosis differs between HCV-GT3 and other genotypes, additional information in HCV-GT1 is necessary. Moreover, data regarding the change of steatosis following successful DAAs in HCV-GT1 were not consistent with conflicting findings among previous studies [12–18].

In this report, approximately 10% of patients had improved steatosis after viral clearance. In contrast, new onset of steatosis appeared in 20% of individuals after SVR and was shown to be independently linked to diabetes and increased BMI in multivariate analysis. Notably, there was no correlation between steatosis and fibrosis alterations in this study, which might be explained by short-term follow-up after DAA treatment and the occurrence of new-onset steatosis in some patients. Indeed, previous studies in Western populations have demonstrated

that weight gain is commonly occur during long-term follow-up after HCV cure with DAAs [43, 44]. For instance, a large prospective study of 11,000 veterans in the U.S. indicated that at least 20% of treated patients had excess weight gain within 2 years of achieving SVR [43]. A recent report from Germany also showed that a substantial weight gain was identified in one-third of HCV-infected patients, particularly among non-obese individuals [44]. Additionally, in the mentioned report the progressive weight gain became more evident during long-term follow-up, while no significant weight change was observed early after completing DAA therapy [44]. Similarly, increased BMI in our patients was particularly observed during long-term follow-up (FUw72) despite there was no significant weight change at FUw24. Based on these data, it is likely that the increased weight might not directly relate to the eradication of HCV. Considering metabolic disturbance is closely related to steatosis development, the increasing steatosis should not be overlooked as it could play a detrimental role after HCV cure. Accordingly, the potential negative impact of steatosis and metabolic derangement that might lead to advanced liver disease in our patients warrants a longer duration of follow-up.

The present study also underlined the influence of *PNPLA3* rs738409 genotype independently associated with progressive steatosis after HCV clearance. In contrast, *TM6SF2* rs58542926 and *MBOAT7* rs641738 were not related to steatosis in our cohort. To our knowledge, this study is the first to examine host genetic factors linked to steatosis progression after HCV eradication following DAA therapy. Indeed, *PNPLA3* rs738409 variant is identified from genome-wide association study as a genetic marker associated with steatosis susceptibility [45]. In a meta-analysis, the pooled results have showed that *PNPLA3* rs738409 is associated with an increased risk of advanced fibrosis and steatosis in chronic HCV infection [46]. Remarkably, emerging data indicate a negative influence of diabetes, obesity and steatosis on the risk of developing HCC and liver-related complications after HCV eradication [19, 47]. Moreover, two cohorts with long-term follow-up in Japanese patients recently demonstrated that *PNPLA3* polymorphism was independently related to future HCC development after SVR [19, 48]. Additionally, a large cohort from Italy showed that combined genetic risk score, including *PNPLA3* variant, was associated with *de novo* HCC in cirrhotic patients treated with DAAs [49]. It should be mentioned that HCV-infected individuals harboring rs738409 G allele have an increased risk of steatosis and HCC according to a meta-analysis [50]. Although the mechanisms by which *PNPLA3* polymorphism links to HCC has not completely known, it is anticipated that this genetic variant might promote hepatocarcinogenesis via metabolic disorders including obesity, diabetes, and steatohepatitis [19]. Together, our data and recent reports indicate that *PNPLA3* polymorphism might be a useful surrogate tool for monitoring the dynamic changes of steatosis, as well as a predictive biomarker for HCC after SVR.

The strength of our study was the sequential use of MRI-based technique, which is now the best non-invasive assessment of fibrosis and steatosis. Unlike previous studies, we prospectively recruited and followed individuals treated with EBR/GZR, thereby minimizing the impact of different DAA regimens and other biases that could be observed in a retrospective design. Despite these advantages, the study had some limitations. First, we included a relatively small number of the HCV/HIV group. Second, the information on insulin resistance, food consumption, lifestyle, physical activity, and alcohol drinking after achieving SVR that might affect steatosis development was inadequate. Finally, another factor that should be considered was the use of ART that might also contribute to developing steatosis in patients with HCV/HIV co-infection. However, the association between ART and steatosis development remains controversial probably due to different study design and statistical analysis among several reports, as mentioned in a recent expert panel review [51]. These limitations might highlight the interaction of several factors that potentially influence the development and progression of steatosis after HCV clearance [52].

In summary, our data revealed that HCV eradication by DAAs was associated with stiffness improvement, as being assessed by MRE. In contrast, increased steatosis based on serial PDFF measurement was observed in a high proportion of responders. Moreover, *PNPLA3* rs738409 non-CC genotype displayed strong association with steatosis progression after SVR. Collectively, the combined clinical parameters and host genetic predictors may allow a better individualized strategy to alleviate progressive steatosis and its adverse clinical outcome among high-risk patients in the era of highly effective DAAs.

## Supporting information

**S1 Fig. Serial MRE and PDFF at each time point in responders and non-responders.** (TIF)

## Acknowledgments

This study was established by Center of Excellence in Hepatitis and Liver Cancer, Department of Biochemistry, Faculty of Medicine, Chulalongkorn University. We would like to thank Mr. Wasan Punyasang (Clinical Epidemiology, Faculty of Medicine, Chulalongkorn University) for the consultation of statistical analysis.

## Author Contributions

**Conceptualization:** Natthaya Chuaypen, Anchalee Avihingsanon, Pisit Tangkijvanich.

**Data curation:** Surachate Siripongsakun, Pantajaree Hiranrat, Natthaporn Tanpowpong, Anchalee Avihingsanon, Pisit Tangkijvanich.

**Funding acquisition:** Pisit Tangkijvanich.

**Investigation:** Natthaya Chuaypen, Natthaporn Tanpowpong, Pisit Tangkijvanich.

**Methodology:** Natthaya Chuaypen.

**Project administration:** Pisit Tangkijvanich.

**Resources:** Surachate Siripongsakun, Pantajaree Hiranrat, Anchalee Avihingsanon.

**Software:** Surachate Siripongsakun, Pantajaree Hiranrat.

**Supervision:** Pisit Tangkijvanich.

**Validation:** Natthaya Chuaypen.

**Visualization:** Natthaya Chuaypen.

**Writing – original draft:** Natthaya Chuaypen.

**Writing – review & editing:** Natthaya Chuaypen, Pisit Tangkijvanich.

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
