## [Decision Letter · Decision Letter 0]

2 Feb 2022

PONE-D-22-01164Long-term improvement of liver fibrosis, but not steatosis, after HCV eradication as assessment by MR-based imaging: Role of metabolic derangement and host genetic variantsPLOS ONE

Dear Dr. Tangkijvanich,

Thank you for submitting your manuscript to PLOS ONE. After careful consideration, we feel that it has merit but does not fully meet PLOS ONE’s publication criteria as it currently stands. Therefore, we invite you to submit a revised version of the manuscript that addresses the points raised during the review process.

We look forward to receiving your revised manuscript.

Kind regards,

Jee-Fu Huang, M.D., Ph.D.

Academic Editor

PLOS ONE

Journal Requirements:

2. We note that you have included the phrase “data not shown” in your manuscript. Unfortunately, this does not meet our data sharing requirements. PLOS does not permit references to inaccessible data. We require that authors provide all relevant data within the paper, Supporting Information files, or in an acceptable, public repository. Please add a citation to support this phrase or upload the data that corresponds with these findings to a stable repository (such as Figshare or Dryad) and provide and URLs, DOIs, or accession numbers that may be used to access these data. Or, if the data are not a core part of the research being presented in your study, we ask that you remove the phrase that refers to these data

Reviewers' comments:

Reviewer's Responses to Questions

5. Review Comments to the Author

Reviewer #1: This manuscript presents some secondary (?) data analysis from a prospective study, which I believe is a randomized clinical trial (?, if the authors can clarify) with a valid NCT number. The study was approved by the respective ethics board. The clinical content is relevant. I have the following questions:

1. Since this is likely a secondary data analysis of data generated from a CT, it is advisable to present some sentences on the appropriate sample size/power the trial was generated on. This is currently missing.

2. Alteration of MRE, PDFF and BMI during follow-up were evaluated using paired t-tests. How was Gaussianity (Normal distribution) of the responses evaluated before applying t-tests? If notmality fails, relevant nonparametric tests are also available.

3. From the design perspective, the data is actually collected longitudinally. So, why was a mixed-effects model not considered, controlling for the relevant covariate effects? Multivariable analysis were conducted; however, it is not clear if the longitudinal (repeated measures) design was factored in the analysis (say, using Proc MIXED in SAS, or something similar).

4. If separate time-points were evaluated (like baseline/time 0 versus time 1, time 1 versus time 2, etc), was multiple comparisons applied, with possible false discovery rate control?

Reviewer #2: In the current study, the authors study the change of fibrosis and steatosis by using MRE and PDFF in CHC patients with/without HIV co-infection who achieved SVR12 after 72 weeks of follow-up period. The authors concluded that HCV eradication was associated with fibrosis improvement. However, progressive steatosis was observed in a proportion of patients, particularly among individuals with metabolic derangement and PNPLA3 variants. There are certain issues to be addressed

Major issue

1. The sample size was so limited that the study of the association of the SNPs with the outcome may turn out to be an incidental finding. Did the SNPs fit the Hardy–Weinberg equilibrium in the population? The discussion should stress on this point.

2. Besides using the cut-off value to perform binary analysis, linear regression analysis should be performed to address the independent factors correlate to the change of liver steatosis since the outcome is quantifiable.

3. Please indicate the reason or reference of using ≥30% or < 30% change of MRE or PDFF as significant change of fibrosis or steatosis. It is critical since it determines the analysis and interpretation of the outcome of interests.

Minor issue

4. Table 1 should include the information of pre-treatment MRE and MRI-PDFF value.

5. The follow-up period was only 1.5 year after DAA in the cohort. The title using the term of “Long-term” is not proper.

6. Did HAART have impact on hepatic steatosis in the cohort

Reviewer #3: The authors aimed to evaluate serial fibrosis and steatosis alterations in patients with HCV genotype 1, who achieved sustained virological response. Fibrosis and steatosis were assessed at baseline, FUw24 and FUw72 by MRE and PDFF, respectively. They concluded that HCV eradication was associated with fibrosis improvement. However, progressive steatosis was observed in a proportion of patients, particularly among individuals with metabolic derangement and PNPLA3 variants. In general, this is an interesting topic and a clearly written paper. However, some issues should be further reconsidered or corrected.

1. Steatosis is recognized as a cofactor influencing the presence and progression of fibrosis in chronic hepatitis C. The authors should evaluate the association of steatosis with fibrosis at baseline and FU. Also, the steatosis grade at baseline and FU should be correlated to the progression of fibrosis.

2. Lack of correlation between steatosis and fibrosis progression should be explained by the short follow-up period after DAA treatment.

3. The authors should compare the serial fibrosis and steatosis alterations between SVR and non-SVR patients. 　　

4. Abstract: archived should be achieved. Some grammatical errors need to be corrected.

5. It is essential that each abbreviation appearing in the abstract or text should be completely described when it was first mentioned such as PNPLA3.

---

## [Author Response · Author response to Decision Letter 0]

16 Apr 2022

Dear the editors

We appreciate the opportunity to submit a revision of our manuscript entitled “Improvement of liver fibrosis, but not steatosis, after HCV eradication as assessment by MR-based imaging: Role of metabolic derangement and host genetic variants” for publication in the PLOS ONE. The funders in this manuscript had no role in study design, data collection and analysis, decision to publish, or preparation of the manuscript. This revised manuscript has been improved with changes made in response to the comments of the reviewers. We have attached a point-by-point response to the comments and have highlighted the changes from the previous version with red color in the revised manuscript. This manuscript is not currently under consideration elsewhere, and all authors have approved the submission of this manuscript for publication in PLOS ONE. Thank you for considering this revised manuscript. We look forward to your kind reply.

Best regards,

Prof. Pisit Tangkijvanich, M.D.

Center of Excellence in Hepatitis and Liver Cancer, Department of Biochemistry, Faculty of Medicine, Chulalongkorn University, Bangkok 10330, Thailand.

Phone (662) – 2564482

E-mail: pisittkvn@yahoo.com

PONE-D-22-01164

Long-term improvement of liver fibrosis, but not steatosis, after HCV eradication as assessment by MR-based imaging: Role of metabolic derangement and host genetic variants

PLOS ONE

Dear Dr. Tangkijvanich,

Thank you for submitting your manuscript to PLOS ONE. After careful consideration, we feel that it has merit but does not fully meet PLOS ONE’s publication criteria as it currently stands. Therefore, we invite you to submit a revised version of the manuscript that addresses the points raised during the review process.

We look forward to receiving your revised manuscript.

Kind regards,

Jee-Fu Huang, M.D., Ph.D.

Academic Editor

PLOS ONE

Journal Requirements:

2. We note that you have included the phrase “data not shown” in your manuscript. Unfortunately, this does not meet our data sharing requirements. PLOS does not permit references to inaccessible data. We require that authors provide all relevant data within the paper, Supporting Information files, or in an acceptable, public repository. Please add a citation to support this phrase or upload the data that corresponds with these findings to a stable repository (such as Figshare or Dryad) and provide and URLs, DOIs, or accession numbers that may be used to access these data. Or, if the data are not a core part of the research being presented in your study, we ask that you remove the phrase that refers to these data

Ans. In this revised manuscript, we have provided these information in Results section (Page 12 and 13). 

Reviewers' comments:

Reviewer's Responses to Questions

5. Review Comments to the Author

Reviewer #1: This manuscript presents some secondary (?) data analysis from a prospective study, which I believe is a randomized clinical trial (?, if the authors can clarify) with a valid NCT number. The study was approved by the respective ethics board. The clinical content is relevant. I have the following questions:

1. Since this is likely a secondary data analysis of data generated from a CT, it is advisable to present some sentences on the appropriate sample size/power the trial was generated on. This is currently missing.

ANS. Thank you very much for the comment. The manuscript was secondary data analyzed from a non-randomized, open-label prospective cohort that was already published (ref#6). The calculated sample size was 95 patients. Allowing for a 5% dropout rate, a total of approximately 100 participants were enrolled.

2. Alteration of MRE, PDFF and BMI during follow-up were evaluated using paired t-tests. How was Gaussianity (Normal distribution) of the responses evaluated before applying t-tests? If notmality fails, relevant nonparametric tests are also available.

3. From the design perspective, the data is actually collected longitudinally. So, why was a mixed-effects model not considered, controlling for the relevant covariate effects? Multivariable analyses were conducted; however, it is not clear if the longitudinal (repeated measures) design was factored in the analysis (say, using Proc MIXED in SAS, or something similar).

4. If separate time-points were evaluated (like baseline/time 0 versus time 1, time 1 versus time 2, etc), was multiple comparisons applied, with possible false discovery rate control?

ANS. We appreciate the comments and would like to response all these questions together regarding the statistical analyses. In this aspect, we have consulted a biostatistician for the analyses (see in Acknowledgement). 

As the reviewer pointed out, a mixed-effects model seems to be an appropriate option and this model has been used in the revised manuscript. Thus, the calculation of MRE, PDFF and BMI during follow-up have been changed to perform repeated measurement analysis with baseline data as covariate using a generalized linear mixed model. Moreover, Post hoc analysis with Bonferroni correction has been used as multiple comparisons. In this revised manuscript, we have provided new P-values in Results (pages 10 and 11). Notably, the results of new P-values are consistent with those of the previously calculated values. We have also added references (new ref#28 and 29) in Statistical analyses (see page 7). 

For univariate and multivariate analyses, we selected the optimal cut-off values of 30% relative changes for MRE and PDFF to estimate the probability of fibrosis and steatosis changes, respectively and the outcomes were assessed by binary analysis. The thresholds of 30% were based on previous reports suggesting accurate endpoints for quantifying dynamic changes of liver fibrosis or steatosis (see new ref#34, 35 and Discussion in page 16). As percentage of BMI change (≥5% from baseline) was considered to be an independent factor associated with an increased PDFF, we further evaluated repeated variables effect on the change in PDFF value using the generalized linear mixed effects models. In this context, our result demonstrated that BMI in separate time-points were not associated with PDFF alteration overtime (P=0.136). (in Results, page 14).

Reviewer #2: In the current study, the authors study the change of fibrosis and steatosis by using MRE and PDFF in CHC patients with/without HIV co-infection who achieved SVR12 after 72 weeks of follow-up period. The authors concluded that HCV eradication was associated with fibrosis improvement. However, progressive steatosis was observed in a proportion of patients, particularly among individuals with metabolic derangement and PNPLA3 variants. There are certain issues to be addressed

Major issue

1. The sample size was so limited that the study of the association of the SNPs with the outcome may turn out to be an incidental finding. Did the SNPs fit the Hardy–Weinberg equilibrium in the population? The discussion should stress on this point. 

ANS. We would like to apologize for not providing this important point in the original manuscript. All PNPLA3, MBOAT7 and TM6SF2 polymorphisms in this study were in Hardy-Weinberg equilibrium (chi-square test: 5.434; P=0.066, 0.348; P=0.840 and 4.275; P=0.118, respectively). In this revised manuscript, we have added a reference (new ref#30) in Statistical analyses (Page 7) and have provided a sentence “In addition, all tested SNPs were in Hardy-Weinberg equilibrium” in Results (Page 12). 

2. Besides using the cut-off value to perform binary analysis, linear regression analysis should be performed to address the independent factors correlate to the change of liver steatosis since the outcome is quantifiable.

3. Please indicate the reason or reference of using ≥30% or < 30% change of MRE or PDFF as significant change of fibrosis or steatosis. It is critical since it determines the analysis and interpretation of the outcome of interests.

ANS. Thank you very much for the suggestion and we would like to response the comments #2 and #3 together. The reason of using ≥30% or < 30% relative changes for MRE or PDFF was based on recent studies (new ref#34 and 35) demonstrating that these thresholds have achieved significant clinical endpoints with high specificity comparing with liver biopsy and could be used as potential surrogate markers for evaluating treatment outcome in clinical trials. For instance, it is recommended that PDFF response defined as ≥30% relative decline in PDFF is associated with ≥1 stage improvement in fibrosis (new ref#34). Thus, in this study we applied the cut-off 30% relative changes in MRE or PDFF values as endpoints for quantifying dynamic changes of liver fibrosis or steatosis, respectively. We have added these data in Discussion (page 16).

Based on these data, we decided to use the same cut-off values for performing binary analysis to identify independent factors associated with significant changes of MRE or PDFF that have clinical relevance. In contrast, linear regression analysis seems to be less suitable for the interpretation because this model does not create the best explanations for the relationship between independent factors and meaningful clinical outcome in terms of MRE or PDFF changes.

Minor issue

4. Table 1 should include the information of pre-treatment MRE and MRI-PDFF value.

ANS. In fact, we already provided these data in the original manuscript using magnetic resonance elastography and proton density fat fraction. In this revised version, we have also added their abbreviations (MRE and PDFF) in Table 1.

5. The follow-up period was only 1.5 year after DAA in the cohort. The title using the term of “Long-term” is not proper.

ANS. We agree with the reviewer’s comment and have changed the title by deleting “Long-term” accordingly.

6. Did HAART have impact on hepatic steatosis in the cohort

ANS. We agree that ART might have an impact on steatosis in the co-infected group. As mentioned in a recent expert panel review (new ref#51), however, the contribution of ART to NAFLD are based on older agents and their associations remain controversial due to the differences in study design and statistical analysis among reports. Moreover, contemporary ART is generally not considered to cause the same metabolic effects as the older agents. We have added this limitation in Discussion (see page 19).

Reviewer #3: The authors aimed to evaluate serial fibrosis and steatosis alterations in patients with HCV genotype 1, who achieved sustained virological response. Fibrosis and steatosis were assessed at baseline, FUw24 and FUw72 by MRE and PDFF, respectively. They concluded that HCV eradication was associated with fibrosis improvement. However, progressive steatosis was observed in a proportion of patients, particularly among individuals with metabolic derangement and PNPLA3 variants. In general, this is an interesting topic and a clearly written paper. However, some issues should be further reconsidered or corrected.

1. Steatosis is recognized as a cofactor influencing the presence and progression of fibrosis in chronic hepatitis C. The authors should evaluate the association of steatosis with fibrosis at baseline and FU. Also, the steatosis grade at baseline and FU should be correlated to the progression of fibrosis.

ANS. Thank you very much for this suggestion. At baseline, we found that there was a weak correlation between steatosis and fibrosis (r=0.262, P=0.017), while there were no association of steatosis and fibrosis at FUw24 and FUw72 (r=0.006, P=0.995 and r=0.101, P=0.362, respectively). These data have already added in the revised manuscript (see Results, page 10). 

2. Lack of correlation between steatosis and fibrosis progression should be explained by the short follow-up period after DAA treatment.

ANS. We have added and explained the relationship of steatosis and fibrosis progression in Discussion (see page 17) as the reviewer’s comment.

3. The authors should compare the serial fibrosis and steatosis alterations between SVR and non-SVR patients. 　　

ANS. We have compared serial fibrosis and steatosis alterations between these groups in Results (see pages 11 and 12) and Supplementary Figure 1. Interestingly, patients with SVR, compared to those without SVR, had comparable MRE values at baseline and FUw24 but had significantly declined levels at FUw72. For serial PDFF values, there was no significant difference between the two groups at baseline, FUw24 and FUw72.

4. Abstract: archived should be achieved. Some grammatical errors need to be corrected.

ANS. We have already corrected this error in Abstract.

5. It is essential that each abbreviation appearing in the abstract or text should be completely described when it was first mentioned such as PNPLA3.

ANS. We apologize for not describing those words in the abstract. In this revised manuscript, the full terms of all abbreviations have been provided.

---

## [Decision Letter · Decision Letter 1]

25 May 2022

Improvement of liver fibrosis, but not steatosis, after HCV eradication as assessment by MR-based imaging: Role of metabolic derangement and host genetic variants

PONE-D-22-01164R1

Dear Dr. Tangkijvanich,

We’re pleased to inform you that your manuscript has been judged scientifically suitable for publication and will be formally accepted for publication once it meets all outstanding technical requirements.

Kind regards,

Jee-Fu Huang, M.D., Ph.D.

Academic Editor

PLOS ONE

Additional Editor Comments (optional):

Reviewers' comments:

Reviewer's Responses to Questions

**Comments to the Author**

1. If the authors have adequately addressed your comments raised in a previous round of review and you feel that this manuscript is now acceptable for publication, you may indicate that here to bypass the “Comments to the Author” section, enter your conflict of interest statement in the “Confidential to Editor” section, and submit your "Accept" recommendation.

Reviewer #1: (No Response)

Reviewer #2: All comments have been addressed

Reviewer #3: All comments have been addressed

---

## [Editor Report · Acceptance letter]

3 Jun 2022

PONE-D-22-01164R1 

Improvement of liver fibrosis, but not steatosis, after HCV eradication as assessment by MR-based imaging: Role of metabolic derangement and host genetic variants 

Dear Dr. Tangkijvanich:

I'm pleased to inform you that your manuscript has been deemed suitable for publication in PLOS ONE. Congratulations! Your manuscript is now with our production department. 

Kind regards, 

on behalf of

Dr. Jee-Fu Huang 

Academic Editor

PLOS ONE